# Estimation of Protein and Amino Acid Requirements in Layer Chicks Depending on Dynamic Model

**DOI:** 10.3390/ani14050764

**Published:** 2024-02-29

**Authors:** Miao Liu, Zhi-Yuan Xia, Hong-Lin Li, Yu-Xuan Huang, Alainaa Refaie, Zhang-Chao Deng, Lv-Hui Sun

**Affiliations:** State Key Laboratory of Agricultural Microbiology, Hubei Hongshan Laboratory, Frontiers Science Center for Animal Breeding and Sustainable Production, College of Animal Sciences and Technology, Huazhong Agricultural University, Wuhan 430070, China; lareina.miao@webmail.hzau.edu.cn (M.L.); zhiyuanxia@webmail.hzau.edu.cn (Z.-Y.X.); lhl011445@webmail.hzau.edu.cn (H.-L.L.); huangyuxuan@webmail.hzau.edu.cn (Y.-X.H.); ainaaeltokhy2001@gmail.com (A.R.)

**Keywords:** Jing Tint 6, layer chicks, amino acids, dynamic model, requirements

## Abstract

**Simple Summary:**

The nutritional requirements of layer hens for protein and amino acids change over time during growth, and traditional feed formulation cannot accurately meet the requirements of the layer hens and tends to waste resources. The use of mathematical models to consider and include the changes in body weight and nutritional requirements of animals at different ages allows appropriate nutrition to be supplied at different stages of animal growth. In this experiment, the Gompertz model was used to consider the growth curve of the layer hens. And the analytical factorization method was used to classify the requirements of the layer hens into growth requirements and maintenance requirements. A dynamic prediction model for protein and amino acid requirements in terms of age was successfully established through a feeding test, slaughter test, nitrogen balance test, and digestive metabolism test. After calculations using the R programming language, it was noticed that the protein requirements of the layer hens decrease with increasing days of age and the amino acid requirements are lower than the recommended values of NRC. The predicted values of the model can provide some reference values for feed formulation in poultry production.

**Abstract:**

Four trials were conducted to establish a protein and amino acid requirement model for layer chicks over 0–6 weeks by using the analytical factorization method. In trial 1, a total of 90 one-day-old Jing Tint 6 chicks with similar body weight were selected to determine the growth curve, carcass and feather protein deposition, and amino acid patterns of carcass and feather proteins. In trials 2 and 3, 24 seven-day-old and 24 thirty-five-day-old Jing Tint 6 chicks were selected to determine the protein maintenance requirements, amino acid pattern, and net protein utilization rate. In trial 4, 24 ten-day-old and 24 thirty-eight-day-old Jing Tint 6 chicks were selected to determine the standard terminal ileal digestibility of amino acids. The chicks were fed either a corn–soybean basal diet, a low nitrogen diet, or a nitrogen-free diet throughout the different trials. The Gompertz equation showed that there is a functional relationship between body weight and age, described as BWt(g) = 2669.317 × exp(−4.337 × exp(−0.019t)). Integration of the test results gave a comprehensive dynamic model equation that could accurately calculate the weekly protein and amino acid requirements of the layer chicks. By applying the model, it was found that the protein requirements for Jing Tint 6 chicks during the 6-week period were 21.15, 20.54, 18.26, 18.77, 17.79, and 16.51, respectively. The model-predicted amino acid requirements for Jing Tint 6 chicks during the 6-week period were as follows: Aspartic acid (0.992–1.284), Threonine (0.601–0.750), Serine (0.984–1.542), Glutamic acid (1.661–1.925), Glycine (0.992–1.227), Alanine (0.909–0.961), Valine (0.773–1.121), Cystine (0.843–1.347), Methionine (0.210–0.267), Isoleucine (0.590–0.715), Leucine (0.977–1.208), Tyrosine (0.362–0.504), Phenylalanine (0.584–0.786), Histidine (0.169–0.250), Lysine (0.3999–0.500), Arginine (0.824–1.147), Proline (1.114–1.684), and Tryptophan (0.063–0.098). In conclusion, this study constructed a dynamic model for the protein and amino acid requirements of Jing Tint 6 chicks during the brooding period, providing an important insight to improve precise feeding for layer chicks through this dynamic model calculation.

## 1. Introduction

The nutritional requirements of animals are a dynamic process that changes with age [1]. For layer chicks, providing appropriate nutrition at different growth stages can promote organ growth and development during the rearing period, subsequently improving their performance during the egg-laying period [2,3]. Jing Tint 6 is a layer breed that has red feathers, has a high egg-laying performance, and produces pink eggs in China. However, conventional feed formulation for layer chicks relies on feeding standards to determine corresponding ratios. This dietary approach will lead to an inadequate nutrient supply during the early stage and excessive nutrient supply in the later stage, which impacts the production potential and the egg-laying performance of Jing Tint 6 chicks [4]. Among various nutrients, dietary protein is digested to release free amino acids, which are absorbed into the bloodstream and utilized by various tissues for synthesizing new proteins [5,6]. Protein deficiency in the diet will lead to reduced growth and even weight loss in chickens [7]. Therefore, establishing a balance between protein and amino acid supply and demand is fundamental and significant in achieving high performance for Jing Tint 6 chicks.

The dynamic model, integrating breed, feed, and environment, can accurately predict nutritional requirements, feed intake, and growth rate of animals at different feeding stages. Previous studies have reported that various non-linear models including Gompertz [8], Logistic [9], Bridges [10], and Richards [11] were applied to describe growth curves and patterns in different animals. Notably, the Gompertz equation (BWt = A × exp [−b × exp(−k × t)]) stood out as the most suitable for characterizing the potential growth curve of poultry [12]. This equation can condense a set of age–weight data into a few parameters, effectively mitigating the impact of experimental errors and fitting well [13,14]. Except for the statistical model, the primary research methods for determining amino acid requirements in poultry encompass the factorial method, gradient addition method, and dietary dilution method [15]. In particular, the factorial method classifies the amino acid requirements of poultry into maintenance requirements and requirements for carcass and feather deposition [16]. These components exhibit a high correlation with the amino acid requirements for bird growth [17]. Previous research has shown that the amino acid requirements determined by the analytical factorization method closely match with actual values recommended by the NRC (1972) for broilers and turkeys [18,19,20,21]. Therefore, the objective of the current study is to determine protein and amino acid requirements by the analytical factorization method and to establish a dynamic nutritional requirement prediction model based on the Gompertz model for commercial layer chicks (Jing Tint 6) during the rearing period. 

## 2. Materials and Methods

### 2.1. Birds, Diets, and Sample Collection 

This study was approved, conducted, and supervised by The Institutional Animal Care and Use Committee of Huazhong Agricultural University, China. 

Trial 1: A total of 90 one-day-old Jing Tint 6 chicks with similar average BWs were selected and randomly allotted to 6 replicates of 15 birds each. The birds were fed a corn–soybean-based diet (Appendix A). The body weight and feed intake of birds were measured weekly. Also, mortality was recorded throughout the trial [22]. Two birds with a close average body weight were selected from each replicate to be weighed and euthanized weekly [23]. The carcasses and feathers were separated, weighed, and collected to determine the protein deposition and amino acid pattern.

Trials 2 and 3: A total of 24 seven-day-old and 24 thirty-five-day-old Jing Tint 6 chicks with similar average BWs were selected. Each group was divided into 2 dietary treatment groups with 4 replicates of 3 birds. The birds were fed a low-nitrogen diet and nitrogen-free diet for one week in trial 2, while they were fed a basal diet and nitrogen-free diet for one week in trial 3. The nitrogen-free diet and low-nitrogen diet (Appendix A) were formulated as in previous studies [12,24]. The first three days were the pre-test period and the last four days were the main test period. Feces from each group were collected during the test period. Briefly, the feces were dried to a constant weight. After feces collection, they were rehydrated indoors for 24 h, weighed, and recorded. Then, they were pulverized, sieved through a 40-mesh sieve, and sealed at −20 °C for further analysis [25,26]. 

Trial 4: A total of 24 ten-day-old and 24 thirty-eight-day-old Jing Tint 6 chicks with similar average BWs were selected. Each group was divided into 2 dietary treatment groups with 4 replicates of 3 birds. The birds were fed a basal diet and nitrogen-free diet for one week where both of the diets were supplemented with 0.5% titanium dioxide (TiO_2_). The birds were euthanized and dissected at 14 and 42 days of age. And then the terminal ileal digesta was collected to determine the standard ileal amino acid digestibility (SID AA).

### 2.2. Dynamic Model Construction

To accurately construct the dynamic models of protein and amino acid requirements in Jing Tint 6 chicks, four trials were conducted. Trial 1 involved the feeding and comparative slaughtering test of chicks, which established relationships between BW and age (t), carcass protein deposition (CPG), and feather protein deposition (FPG), respectively, which are expressed as
BW = f(t), CP = f(BW), FP = f(BW), PRc = CPG = f(t), and PRf = FPG = f(t)
where CP is carcass protein absolute quantity, FP is feather protein absolute quantity, PRc is carcass growth requirements, and PRf is feather growth requirements. The expression of the Gompertz equation is described as
BWt = A × exp[−b × exp(−k × t)](1)
where t denotes the age, BWt is the body weight at age t, k is the average growth rate, A is the mature BW, b is the conditioning parameter, and exp is the bottom of the logarithm of Bairan. The amino acid patterns of feather and carcass proteins (AAf and AAc) were also determined in trial 1. Trial 2 was a nitrogen balance test, which determined the protein maintenance requirements (PRm) and the amino acid pattern for maintenance requirements (AAm). Trials 3 and 4 were digestion and metabolism tests, which determined net protein utilization (NPU) and the standard terminal ileal digestibility of amino acids (SID AA). Combining the parameters determined from these four trials, the total protein requirement (PR) is described as
PR = (PRm + PRc + PRf)/NPU = (c × BW^0.75^ + CPG + FPG)/NPU(2)
and the total amino acid requirement (AAR) is described as
AAR = (PRm × AAm)/SID AA +(PRc × AAc)/NPU + (PRf × AAf)/NPU(3)

This dynamic predictive model provides an insight into daily age-based protein and amino acid nutritional requirements for Jing Tint 6 chicks during the rearing period. 

### 2.3. Protein, Amino Acid, and Creatinine Analyses

Crude protein in the feed, feathers, carcasses, chyme, and feces was determined with a Kjeldahl nitrogen meter [27]. Tryptophan was determined by the national standard GBT15400-2018 method; other amino acids in the feed, feathers, carcasses, digesta, and feces were determined by a fully automated amino acid analyzer [28]. Creatinine content in the feces from trial 2 was analyzed with a Sarcosine oxidase method using its specific assay kits (C011-2-1) purchased from the Nanjing Jiancheng Bioengineering Institute of China. TiO_2_ in feed and ileal digesta from trial 4 was measured by the spectrophotometric method [29]. 

### 2.4. Statistical Analyses

The non-linear regression method in SPSS 26.0 software was used throughout the experiment to fit the equation and to determine the optimal estimates of the parameters of the model and the function. Data were analyzed by the independent samples t-test or analysis of variance (ANOVA) in SPSS 26.0 software, and multiple comparisons were made using Duncans. Data are presented as mean ± SE, and the level of significance was set at *p* < 0.05. 

## 3. Results

### 3.1. Fitting the Model of Growth in Jing Tint 6 Chicks by Gompertz Equation 

Based on the BW data of Jing Tint 6 chicks at different ages in trial 1 (Appendix A), the model of growth curves was fitted by the Gompertz equation. Notably, the multiple correlation coefficient (R^2^) of this model is 0.9996, indicating that the Gompertz equation can accurately predict the growth process of the layer chicks. After determining the relevant parameters of the Gompertz equation, the functional relationship between the BW and age of Jing Tint 6 chicks is described as
BWt(g) = 2669.317 × exp[−4.337 × exp(−0.019 × t)](4)

### 3.2. Fitting the Model of Carcass and Feather Protein Deposition in Jing Tint 6 Chicks 

Based on the corresponding body composition of Jing Tint 6 chicks at different ages in trial 1 (Appendix A), four different mathematical equations were applied to fit the relationship between carcass/feather protein and BW of Jing Tint 6 chicks. As shown in Table 1, the model coefficient of Y = b_0_ + b_1_ × X + b_2_ × X^2^ (R^2^ = 0.999) is higher than that of other models in carcass protein, while the model coefficient of Y = b_0_ × exp(b_1_ × X) (R^2^ = 0.999) is higher than that of other models in feather protein. Therefore, after determining the relevant parameters, the relationship between carcass protein and BW is described as
Y = −2.85 + 0.192 × X − 5.581 × 10 − 5 × X^2^(5)
and the relationship between feather protein and BW is described as
Y = 1.248 × exp(0.007 × X)(6)

The growth curve equations were substituted into the above-mentioned two mathematical models and each equation was differentiated separately to establish the dynamic functional relationship between weight gain, carcass protein deposition (CPG), feather protein deposition (FPG), and age of Jing Tint 6 chicks, described as
Gt(g) = 219.96 × exp(−0.019 × t − 4.337 × exp(−0.019 × t))(7)
FPGt(g) = 1.92157 × exp(−0.019 × t + 18.6852 × exp(−4.337 × exp(−0.019 × t))) − 4.337 × exp(−0.019 × t)(8)
CPGt(g) = exp(−0.019 × t) × (42.2323 × exp(−4.337 × exp(−0.019 × t))) − 65.5368 × exp(−8.674 × exp(−0.019 × t))(9)
where “t” is age, “Gt” is weight gain at different ages, “CPGt” is carcass protein deposition at different ages, and “FPGt” is feather protein deposition at different ages. 

### 3.3. Amino Acid Patterns in Carcasses and Feathers of Jing Tint 6 Chicks

Except for Valine, Methionine, Lysine, and Tryptophan, there was no significant difference in the amino acid composition of the carcass at 3–6 weeks (*p* > 0.05). However, there were significant differences between amino acid compositions at 0–2 weeks (*p* < 0.05), mainly manifested in the difference between day 0 and days 7 and 14, which may be because the chicks on day 0 were affected by the remaining yolk in the body after birth [30]. There were significant differences in feathers overall (*p* < 0.05), which may be because the ratio of quill-coverts to contour feathers had changed with age, and different parts of the same feather had different amino acid compositions of keratin, which caused the difference [31] (Appendix A). In this experiment, the growth of laying hens was divided into two stages of 0–2 weeks and 3–6 weeks. Therefore, the amino acid composition patterns of carcasses and feathers were expressed using 0 to 2 weeks (average of 0, 7, and 14 days) and from 3 to 6 weeks (average of 21, 28, 35, and 42 days), respectively, and were described as AAc1, AAc2 and AAf1, AAf2, respectively (Table 2).

### 3.4. Protein and Amino Acid Maintenance Requirements of Jing Tint 6 Chicks

The results of nitrogen balance trials of Jing Tint 6 chicks are shown in Table 3. The endogenous nitrogen excretion in the second week and sixth week of feeding nitrogen-free diets was 18.03 mg/d and 34.47 mg/d, respectively. In the maintenance state, the loss of nitrogen from feathers, dander, and other skin coverings in the low-nitrogen diets was 48.89 mg/d and 68.53 mg/d, and the excretion of creatinine was 0.25 mg/d and 0.42 mg/d. Therefore, the protein maintenance requirements of Jing Tint 6 chicks during 0–2 weeks and 3–6 weeks are described as PRm = 14.58 × BW^0.75^ and PRm = 10.52 × BW^0.75^, respectively. Based on the daily endogenous nitrogen excretion, body surface nitrogen loss, amino acid and creatinine excretion, and the amino acid composition pattern of carcass and feather proteins of Jing Tint 6 chicks, the amino acid patterns were calculated during 0 to 2 weeks (Table 4) and 3 to 6 weeks (Table 5) and described as AAm1 and AAm2, respectively. The nitrogen excreted in the form of amino acids accounts for about a quarter of the endogenous nitrogen, which shows that a large number of amino acids cannot be used for protein synthesis during the protein turnover process in the body. However, they are oxidized and decomposed to be excreted in the form of urinary nitrogen.

### 3.5. Net Protein Utilization and Standard Ileal Terminal Digestibility of Amino Acids at Different Growth Stages

As shown in Table 6, the net protein utilization of Jing Tint 6 chicks at 14 and 42 days of age was 76.13% and 78.99%, respectively. The net protein utilization rate was slightly lower in the early stage than in the late stage, which might be due to the imperfect organ development in the early life stages of Jing Tint 6 chicks. As shown in Table 7, the standard terminal ileal digestibility of amino acids was relatively stable, indicating that Jing Tint 6 chicks had better digestion and absorption of amino acids; it was described as SIDAA1 and SIDAA2 at the second and sixth weeks, respectively.

### 3.6. Dynamic Model of Protein and Amino Acid Requirements

The test divided the protein and amino acid requirements into two parts, growth requirements and maintenance requirements. The protein requirements of Jing Tint 6 chicks during 0–2 weeks and 3–6 weeks were expressed as PR1 and PR2, respectively, and the amino acid requirements were referred to as AAR1 and AAR2, respectively. Overall, the total equations for protein and amino acid requirements of Jing Tint 6 chicks are described as
PR1 = PRm1 + PRc1 + PRf1 = ((exp(−0.019 × t) × (42.2323 × exp(−4.337 × exp(−0.019 × t))) − 65.5368 × exp(−8.674 × exp(−0.019 × t)) + 1.92157 × exp((−0.019 × t + 18.6852 × exp(−4.337 × exp(−0.019 × t))) − 4.337 × exp(−0.019 × t)) + 5.4145 × (exp(−4.337 × exp(−0.019 × t)))^0.75^)/76.13%(10)
PR2 = PRm2 + PRc2 + PRf2 = ((exp(−0.019 × t) × (42.2323 × exp(−4.337 × exp(−0.019 × t))) − 65.5368 × exp(−8.674 × exp(−0.019 × t)) + 1.92157 × exp((−0.019 × t + 18.6852 × exp(−4.337 × exp(−0.019 × t))) − 4.337 × exp(−0.019 × t)) + 3.9068 × (exp(−4.337 × exp(−0.019 × t)))^0.75^)/78.99%(11)
AAR1 = PRm1 × AAm1 + PRc1 × AAc1 + PRf1 × AAf1 = 5.4145 × (exp(−4.337 × exp(−0.019 × t)))^0.75^ × AAm1/83.49% + 1.92157 × exp((−0.019 × t + 18.6852 × exp(−4.337 × exp(−0.019 × t))) − 4.337 × exp(−0.019 × t)) × AAc1/76.13% + (exp(−0.019 × t) × (42.2323 × exp(−4.337 × exp(−0.019 × t))) − 65.5368 × exp(−8.674 × exp(−0.019 × t)) × AAf1/76.13%(12)
AAR2 = PRm2 × AAm2 + PRc2 × AAc2 + PRf2 × AAf2 = 3.9068 × (exp(−4.337exp(−0.019 × t)))^0.75^ × AAm2/83.17% + 1.92157 × exp((−0.019 × t + 18.6852 × exp(−4.337 × exp(−0.019 × t))) − 4.337 × exp(−0.019 × t)) × AAc2/78.99% + (exp(−0.019 × t) × (42.2323 × exp(−4.337 × exp(−0.019 × t))) − 65.5368 × exp(−8.674 × exp(−0.019 × t))) × AAf2/78.99%(13)

The equations were run through the R language to obtain the protein and amino acid requirements of Jing Tint 6 chicks (Table 8). The comparison of the protein and amino acid nutritional requirements of the tested Jing Tint 6 chicks with the feeding standard of Jing Tint 6 chicks showed that the protein requirements generally show a correlation with weekly age, and the protein requirements for the growth of Jing Tint 6 chicks gradually decrease with increasing weekly age as the protein requirements in weeks 1 and 2 were higher than the standard, while the protein requirements of 3, 4, 5, and 6 weeks were lower than the standard. It is also shown that amino acid requirements were all below standard requirements as Threonine requirements were close to standard values and gradually decreasing, while Methionine and Lysine requirements were close to half of the standard values.

## 4. Discussion

Under the traditional feeding mode, producers tend to supplement animals with nutrients beyond their requirements to ensure optimal growth performance, resulting in wasting feed resources, higher production costs, and increasing environmental pollution [32,33]. In this study, we established a dynamic protein and amino acid nutrient requirement model for layer chicks at different growth stages, providing them with the appropriate nutrients to meet their growth and production needs. Consequently, this contributed to more scientifically and rationally formulated feed and reduced resource wastage in poultry production. The protein requirements of Jing Tint 6 chicks from 0 to 6 weeks were calculated by modeling as 21.15, 20.54, 18.26, 18.77, 17.79, and 16.51, respectively. The comparison with the feeding standard of “Jing Tint 6” revealed higher model-predicted protein requirements from 0 to 2 weeks. This discrepancy may be attributed to the rapid growth and development of the layer chicks and their robust metabolism. However, in the early stage, their digestive system is not fully developed, and they have a small stomach volume with poor muscular stomach grinding ability. Consequently, they exhibit higher nutritional requirements and necessitate the consumption of high-quality protein feeds [34]. Previous research showed that chicks achieved optimal body weight when fed a 22% protein diet [35]. The protein requirements of chicks gradually declined from 3 to 6 weeks, which is a period characterized by more stable carcass growth and an increased proportion of feather growth.

Jing Tint 6 chicks require ten essential amino acids including Methionine, Lysine, Threonine, Isoleucine, Leucine, Valine, Tryptophan, Arginine, Histidine, and Phenylalanine. Among these, Lysine, Methionine, and Threonine are identified as the three most limited amino acids for Jing Tint 6 chicks [36]. Notably, the recommended content of Threonine, Methionine, and Lysine in the feeding standard for “Jing Tint 6” is 0.73–0.75%, 0.45–0.46%, and 1.00–1.10%, respectively. In this study, the Threonine, Methionine, and Lysine requirements for Jing Tint 6 chicks from 0 to 6 weeks were calculated as 0.601–0.75%, 0.210–0.267%, and 0.399–0.500%, respectively. The calculated Threonine requirement value was consistent with the feeding standard of “Jing Tint 6” and a previous study [37]. In comparison, NRC (1994) suggested that the Methionine value was 0.28% (0–6 weeks of age). Notably, the model-predicted Methionine value closely matched the NRC (1994) results and was lower than the feeding standards and other studies [38,39]. The model-predicted Lysine requirement value was below the NRC (1972)-recommended value of 0.80% for Lysine (0–6 weeks of age) and the Practical Poultry Nutrition for Canada-recommended value of 1.10% for Lysine (0–5 weeks of age) [40]. Lysine requirements predicted by the model were lower than all the aforementioned criteria and the results reported in other studies [41,42]. The variation in results may have arisen from substantial differences among various breeds of Jing Tint 6 chicks in terms of body weight, daily weight gain, and production performance. These disparities contribute to variations in the requirements and proportions of certain amino acids used for maintenance and weight gain [43]. Slower-growing breeds generally exhibit lower demand for amino acids, resulting in reduced requirements for protein and Methionine [44].

The model-predicted requirements for Tryptophan fell within the range of the lowest and highest recommended levels from other studies [45,46]. Valine is slightly higher than the results of previous studies [47]. Lelis et al. suggested that increasing the digestible Valine in the diet can linearly enhance the feed intake of Jing Tint 6 chicks, justifying a moderate increase in Valine [48]. Arginine is a precursor to protein, creatine, Proline, polyamines, and nitric oxide, and it can help improve production performance and immunity in Jing Tint 6 chicks [49]; the model-predicted requirements for Arginine were slightly higher than the results from previous studies but remained within an acceptable range without causing adverse effects [50]. Isoleucine levels below 4.0 g/kg [51] and above 8.1 g/kg [52] were detrimental to Jing Tint 6 chicks’ feed intake, but the model predicted Isoleucine requirements to fall between these limits.

In general, the model predictions generally align with previous studies. Any discrepancies may arise from the dynamic changes in protein increase and chemical composition within poultry during the growth process, thereby influencing the model’s assessment of amino acid nutritional requirements. Additionally, the variation in the growth rates of different layer hen breeds contributes to distinct protein and amino acid requirements reflecting breed-specific characteristics. To validate the accuracy of the model, further experiments will be conducted.

## 5. Conclusions

In this study, a dynamic prediction model of protein and amino acid requirements of Jing Tint 6 chicks during the brooding period was established. The weekly protein and amino acid requirements could be accurately calculated through this model, which contributed to the precise preparation of feed for layer chicks, thereby promoting the efficient production of layer hens. This model also provided an important reference for assessing the nutritional requirements of other animals, which will help improve production efficiency throughout the animal industry.

## Figures and Tables

**Table 1 animals-14-00764-t001:** Values of model parameters for different mathematical models fitting the relationship between carcass or feather protein and body weight in Jing Tint 6 chicks ^1^.

Ingredients	Equation Expression	Model Parameter				
		b_0_	b_1_	b_2_	R^2^	SE
carcass	Y = b_0_ + b_1_X	−1.283	0.169		0.998	0.689
	Y = b_0_ + b_1_X + b_2_X^2^	−2.850	0.192	−5.581 × 10^−5^	0.999	0.781
	Y = b_0_(X^b1^)	0.141	1.027		0.997	0.940
	Y = b_0_exp(b_1_X)	10.662	0.005		0.926	2.457
feather	Y = b_0_ + b_1_X	−2.568	0.053		0.895	1.752
	Y = b_0_ + b_1_X + b_2_X^2^	2.312	−0.019	0.000	0.992	0.876
	Y = b_0_(X^b1^)	0.000	1.965		0.976	0.221
	Y = b_0_exp(b_1_X)	1.248	0.007		0.999	0.076

^1^ “X” is the body weight; “Y” is the amount of carcass or feather protein.

**Table 2 animals-14-00764-t002:** Amino acid composition in carcass and feather proteins (% of N × 6.25) of Jing Tint 6 chicks ^1^.

Amino Acid	Feather		Carcass	
	0–2 Weeks	3–6 Weeks	0–2 Weeks	3–6 Weeks
Aspartic acid	5.89 ± 0.20	5.70 ± 0.20	7.32 ± 0.20	6.08 ± 0.06
Threonine	3.47 ± 0.12	3.59 ± 0.22	4.01 ± 0.11	3.52 ± 0.07
Serine	7.64 ± 0.05	7.67 ± 0.47	4.31 ± 0.21	3.47 ± 0.05
Glutamic acid	8.63 ± 0.02	8.85 ± 0.41	12.42 ± 0.24	11.29 ± 0.15
Glycine	5.59 ± 0.18	5.25 ± 0.28	7.23 ± 0.12	6.84 ± 0.09
Alanine	4.16 ± 0.01	4.40 ± 0.56	7.63 ± 0.11	6.78 ± 0.10
Valine	5.43 ± 0.15	5.58 ± 0.37	4.17 ± 0.12	3.26 ± 0.30
Cystine	6.82 ± 0.19	7.29 ± 0.25	2.01 ± 0.12	1.87 ± 0.12
Methionine	0.84 ± 0.03	0.81 ± 0.03	2.63 ± 0.04	2.60 ± 0.04
Isoleucine	3.34 ± 0.04	3.57 ± 0.06	3.71 ± 0.09	3.46 ± 0.04
Leucine	5.60 ± 0.03	5.73 ± 0.28	6.61 ± 0.16	6.01 ± 0.12
Tyrosine	2.37 ± 0.38	2.19 ± 0.35	2.67 ± 0.10	2.14 ± 0.06
Phenylalanine	3.69 ± 0.11	3.54 ± 0.11	3.80 ± 0.08	3.36 ± 0.06
Histidine	0.93 ± 0.05	0.58 ± 0.05	2.23 ± 0.07	2.77 ± 0.08
Lysine	1.55 ± 0.15	1.62 ± 0.15	5.32 ± 0.17	4.84 ± 0.09
Arginine	5.42 ± 0.18	5.09 ± 0.37	5.42 ± 0.13	4.75 ± 0.10
Proline	8.25 ± 0.08	8.27 ± 0.41	5.65 ± 0.04	4.38 ± 0.15
Tryptophan	0.42 ± 0.09	0.27 ± 0.07	0.73 ± 0.04	0.63 ± 0.02

^1^ Values are means ± SE, n = 6. Means in a row with different superscripts are different, *p* < 0.05.

**Table 3 animals-14-00764-t003:** Results of nitrogen balance trials in Jing Tint 6 chicks ^1^.

Item	Week 2–3		Week 5–6	
	Low-Nitrogen Group	Non-Nitrogen Group	Low-Nitrogen Group	Non-NitrogenGroup
Initial body weight, g	105.48 ± 4.91	102.54 ± 7.38	377.08 ± 4.38	377.92 ± 3.80
Final body weight, g	70.14 ± 5.17	83.55 ± 8.34	305.13 ± 6.56	284.83 ± 8.88
Feed intake, g/day	7.96 ± 0.22	5.24 ± 0.39	17.7 ± 2.10	11.30 ± 1.50
N intake, mg/day	75.22 ± 0.002	5.16 ± 0.0004	166.82 ± 0.019	11.12 ± 0.002
N excretion, mg/day	26.33 ± 0.007	18.03 ± 0.002	98.28 ± 0.011	34.47 ± 0.004
N retention, mg/day	48.89 ± 0.007		68.53 ± 0.03	
Creatinine excretion, mg/day	0.25 ± 0.04		0.42 ± 0.12	
Parameter “c”	14.58		10.52	

^1^ Values are means ± SE, n = 4.

**Table 4 animals-14-00764-t004:** Amino acid partition, pattern, and requirements for maintenance in Jing Tint 6 chicks during weeks 0–2 ^1^.

Item	Endogenous Amino Acid Excretion			Loss of Feather Dander (mg/d)	Total Losses (mg/d)	Amino Acid Maintenance Mode (%)
	Amino Acid Pattern (mg/d)	Non-Amino Acid Pattern (mg/d)	Creatinine Pattern (mg/d)			
Nitrogen	4.38 ± 1.12	13.56 ± 1.22	0.093 ± 0.02	48.89 ± 0.01	66.92	
Aspartic acid	2.71 ± 0.62	6.20 ± 0.51		18.00 ± 2.86	26.91	6.43
Threonine	1.61 ± 0.42	3.40 ± 0.28		10.60 ± 1.68	15.61	3.73
Serine	2.13 ± 0.74	3.65 ± 0.29		23.35 ± 3.70	29.13	6.96
Glutamic acid	4.62 ± 1.20	10.52 ± 0.87		26.37 ± 4.18	41.51	9.93
Glycine	2.54 ± 0.71	6.13 ± 0.51	0.17 ± 0.03	17.08 ± 2.71	25.92	6.20
Alanine	2.19 ± 0.71	6.47 ± 0.53		12.71 ± 2.02	21.36	5.11
Valine	1.69 ± 0.47	3.53 ± 0.29		16.59 ± 2.63	21.81	5.22
Cystine	2.54 ± 0.57	1.70 ± 0.14		20.84 ± 3.31	25.08	6.00
Methionine	0.88 ± 0.30	2.23 ± 0.18	0.33 ± 0.06	2.57 ± 0.41	6.00	1.44
Isoleucine	1.22 ± 0.39	3.14 ± 0.26		10.21 ± 1.62	14.57	3.48
Leucine	2.10 ± 0.75	5.60 ± 0.46		17.11 ± 2.72	24.81	5.93
Tyrosine	0.55 ± 0.18	2.26 ± 0.19		7.24 ± 1.15	10.05	2.40
Phenylalanine	1.50 ± 0.36	3.22 ± 0.27		11.28 ± 1.79	16.00	3.83
Histidine	0.73 ± 0.13	1.89 ± 0.16		2.84 ± 0.45	5.46	1.31
Lysine	1.24 ± 0.10	4.51 ± 0.37		4.74 ± 0.75	10.48	2.51
Arginine	1.49 ± 0.44	4.59 ± 0.38	0.39 ± 0.07	16.56 ± 2.63	23.04	5.51
Proline	2.23 ± 0.75	4.79 ± 0.39		25.21 ± 4.00	32.22	7.70
Tryptophan	0.36 ± 0.04	0.62 ± 0.05		1.28 ± 0.20	2.26	0.54

^1^ Values are means ± SE, n = 4.

**Table 5 animals-14-00764-t005:** Amino acid partition, pattern, and requirements for maintenance of Jing Tint 6 chicks during weeks 3–6 ^1^.

Item	Endogenous Amino Acid Excretion			Loss of Feather Dander (mg/d)	Total Losses (mg/d)	Amino Acid Maintenance Mode (%)
	Amino Acid Pattern (mg/d)	Non-Amino Acid Pattern (mg/d)	Creatinine Pattern (mg/d)			
Nitrogen	8.64 ± 1.34	25.67 ± 2.06	0.16 ± 0.04	68.53 ± 0.03	103.00	
Aspartic acid	6.56 ± 0.71	9.75 ± 0.88		24.42 ± 1.95	40.73	6.33
Threonine	3.66 ± 0.35	5.65 ± 0.51		15.38 ± 1.23	24.69	3.84
Serine	4.05 ± 0.33	5.57 ± 0.50		32.85 ± 2.62	42.47	6.60
Glutamic acid	9.22 ± 1.13	18.11 ± 1.64		37.91 ± 3.03	65.24	10.13
Glycine	4.63 ± 0.59	10.97 ± 0.99	0.28 ± 0.08	22.49 ± 1.80	38.37	5.96
Alanine	5.60 ± 0.69	10.88 ± 0.98		18.85 ± 1.51	35.32	5.49
Valine	4.18 ± 0.43	5.23 ± 0.47		23.90 ± 1.91	33.31	5.17
Cystine	4.81 ± 0.23	3.00 ± 0.27		31.23 ± 2.49	39.03	6.06
Methionine	1.62 ± 0.29	4.17 ± 0.38	0.56 ± 0.16	3.64 ± 0.25	9.82	1.53
Isoleucine	2.90 ± 0.39	5.55 ± 0.50		15.29 ± 1.22	23.75	3.69
Leucine	4.38 ± 052	9.64 ± 0.87		24.54 ± 1.96	38.56	5.99
Tyrosine	1.46 ± 0.19	3.43 ± 0.31		9.38 ± 0.75	14.28	2.22
Phenylalanine	3.37 ± 0.22	5.39 ± 0.49		15.16 ± 1.21	23.93	3.72
Histidine	0.84 ± 0.08	4.44 ± 0.40		2.48 ± 0.20	7.77	1.21
Lysine	2.71 ± 0.22	7.76 ± 0.70		6.94 ± 0.55	17.41	2.70
Arginine	2.51 ± 0.19	7.62 ± 0.69	0.66 ± 0.19	22.46 ± 1.74	32.59	5.06
Proline	5.91 ± 0.80	7.03 ± 0.63		35.42 ± 2.83	48.36	7.51
Tryptophan	0.70 ± 0.09	1.01 ± 0.09		1.16 ± 0.09	2.86	0.44

^1^ Values are means ± SE, n = 4.

**Table 6 animals-14-00764-t006:** Results of a nitrogen balance trial of layer chicks consuming a protein-containing diet ^1^.

Item	Week 2		Week 6	
	Standard Diet Group 1	Non-Nitrogen Diet Group 1	Standard Diet Group 2	Non-Nitrogen Diet Group 2
Initial body weight, g	100.80 ± 2.91	99.78 ± 2.36	275.85 ± 3.10	273.82 ± 5.14
Feed intake, g/day	11.57 ± 0.76	5.82 ± 0.31	19.33 ± 0.84	9.96 ± 0.42
Feces and urine excretion, g/day	2.38 ± 0.17	0.21 ± 0.01	3.98 ± 0.10	0.44 ± 0.03
Protein intake, g/day	2.22 ± 0.15	-	3.74 ± 0.16	-
Protein excretion, g/day	0.60 ± 0.06	0.065 ± 0.01	0.96 ± 0.05	0.17 ± 0.02
Net protein availability, %	76.13 ± 1.99		78.99 ± 0.70	

^1^ Values are means ± SE, n = 4.

**Table 7 animals-14-00764-t007:** Standard ileal amino acid digestibility of Jing Tint 6 chicks at day 14 and 42 ^1^.

Amino Acid	14 Days Old	42 Days Old
Aspartic acid	85.91 ± 1.70	87.22 ± 0.86
Threonine	80.56 ± 2.11	81.62 ± 2.20
Serine	84.18 ± 2.34	86.12 ± 2.50
Glutamic acid	89.08 ± 1.76	90.60 ± 1.47
Glycine	76.66 ± 2.45	78.61 ± 1.97
Alanine	81.69 ± 2.52	81.54 ± 2.57
Valine	78.45 ± 2.68	81.07 ± 3.40
Cystine	87.96 ± 1.78	87.85 ± 3.27
Methionine	85.43 ± 3.48	85.25 ± 2.67
Isoleucine	83.05 ± 2.29	83.44 ± 2.25
Leucine	84.90 ± 2.40	87.33 ± 2.45
Tyrosine	72.35 ± 3.66	73.54 ± 2.77
Phenylalanine	86.02 ± 1.61	83.96 ± 2.60
Histidine	83.29 ± 2.61	79.75 ± 2.94
Lysine	72.96 ± 4.90	81.09 ± 2.48
Arginine	86.73 ± 3.06	87.80 ± 2.08
Proline	87.92 ± 2.98	90.74 ± 1.57
Tryptophan	87.26 ± 0.45	77.88 ± 2.30

^1^ Values are means ± SE, n = 4.

**Table 8 animals-14-00764-t008:** Ratio of protein and amino acids to feed (%) predicted by the model.

Item	Week 1	Week 2	Week 3	Week 4	Week 5	Week 6
Protein	21.15	20.54	18.26	18.77	17.79	16.51
Aspartic acid	1.284	1.251	1.103	1.111	1.059	0.992
Threonine	0.750	0.730	0.684	0.686	0.650	0.601
Serine	1.542	1.489	1.333	1.301	1.171	0.984
Glutamic acid	1.925	1.880	1.755	1.782	1.726	1.661
Glycine	1.227	1.196	1.041	1.059	1.028	0.992
Alanine	0.961	0.943	0.909	0.933	0.921	0.912
Valine	1.121	1.086	0.996	0.980	0.894	0.773
Cystine	1.347	1.296	1.235	1.193	1.050	0.843
Methionine	0.226	0.225	0.210	0.226	0.241	0.267
Isoleucine	0.715	0.695	0.675	0.676	0.639	0.590
Leucine	1.208	1.176	1.093	1.099	1.046	0.977
Tyrosine	0.504	0.490	0.412	0.413	0.391	0.362
Phenylalanine	0.786	0.764	0.670	0.671	0.634	0.584
Histidine	0.229	0.226	0.169	0.189	0.212	0.250
Lysine	0.414	0.412	0.399	0.429	0.455	0.500
Arginine	1.147	1.114	0.951	0.951	0.896	0.824
Proline	1.684	1.629	1.463	1.435	1.302	1.114
Tryptophan	0.098	0.096	0.063	0.067	0.069	0.072

## Data Availability

The original contributions presented in the study are included in the article/Appendix A.

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
