# Peer review of "Estimation of Protein and Amino Acid Requirements in Layer Chicks Depending on Dynamic Model"

_animals, 2024, doi:10.3390/ani14050764_

Round 1

Reviewer 1 Report

Comments and Suggestions for Authors

The authors made a valuable and interesting work, bringing new knowledge in the estimation of nutritional requirements for poultry.

Some of my comments are next:

Line 9: nutritional requirements instead of nutritional needs, please correct it in all cases. -Requirements- is the typically word in nutrition to define the needs of nutrients in some species. Please use it in whole document.

Line 11, 13: -to fit the- what you mean? To keep? To achieve? it causes a confusion….

Line 22: 90 day old Jing Tint 6 chicks.. Please use the same words in other cases

Line 74: NRC 1972 is the latest recomamandation for nutrients requirements of Jing Tint 6 birds? Please check

Lie 86-87: please use scientific technical words

Throughout the manuscript, certain words that are not technically and scientifically correct must be corrected, also the English language would need improvements.

The conclusion must be improved and should have some phrases with the technological transfer and impact into the practice of the field.

Why the authors consider necessary to estimate the nutritional requirements for Jing Tint 6 chicks ? Why this variety?

Comments on the Quality of English Language

The english language must be improved and the authors must to use technical and scientific words from the field.

Author Response

Reviewer #1: The authors made a valuable and interesting work, bringing new knowledge in the estimation of nutritional requirements for poultry. Some of my comments are next:

1. Line 9: nutritional requirements instead of nutritional needs, please correct it in all cases. -Requirements- is the typically word in nutrition to define the needs of nutrients in some species. Please use it in whole document.

Response: Thank you for pointing out this. We have corrected “needs” to “requirements” in whole document. Please check them throughout the manuscript. Thanks.

2. Line 11, 13: -to fit the- what you mean? To keep? To achieve? it causes a confusion….

Response: Thank you for pointing out this. We have corrected “to fit the” to “to consider and include” in the Line 11, and have corrected “to fit the” to “to consider” in the Line 14.

3. Line 22: 90 day old Jing Tint 6 chicks. Please use the same words in other cases Response: Thank you for pointing out this. “90” is the number of chicks not the day old of chicks. We have revised “90 1-d-old” to “90 one-day-old” in Line 24, and have revised the same description in other cases to make clearer. Please check them. Thanks.

4. Line 74: NRC 1972 is the latest recommendation for nutrients requirements of Jing Tint 6 birds? Please check

Response: Thank you for pointing out this. The NRC has not issued feeding standards for Jing Tint 6 birds, because Jing Tint 6 bird, a new breed in China, is bred by China Agricultural University and Beijing Huadu Yukou Poultry Industry Co., Ltd. In Lines 75-77, we reviewed serval previous researches related to this study, and “NRC 1972” here is not for Jing Tint 6 birds. Thanks for your understanding.

5. Line 86-87: please use scientific technical words

Response: Thank you for pointing out this. We have revised this sentence as follows: Two birds with a close average body weight were selected from each replicate to be weighed and euthanized weekly. Please check Lines 89-91, thanks.

6. Throughout the manuscript, certain words that are not technically and scientifically correct must be corrected, also the English language would need improvements.

Response: Thank you for pointing out this. We have asked a native English speaker, who has helped us revised the English language, and have revised some words and sentences throughout the manuscript to make more technical and scientific. Please check them in highlighted text, thanks.

7. The conclusion must be improved and should have some phrases with the technological transfer and impact into the practice of the field.

Response: Thank you for pointing out this. We have revised the sentences in the conclusion. Please check the Lines 317-321, thanks.

8. Why the authors consider necessary to estimate the nutritional requirements for Jing Tint 6 chicks? Why this variety?

Response: Thank you for pointing out this. This is a great comment. Jing Tint 6 is a layer breed with red feathers, high egg-laying, and producing pink eggs in China. Compare to other breeds of layer hen, Jing Tint 6 has higher economic value because of its high production performance. However, conventional feed formulation for layer chicks relies on feeding standards to determine corresponding ratios. This dietary approach will lead to an inadequate nutrient supply during the early stage and excessive nutrient supply in the later stage, which impacts the production potential and the egg-laying performance of Jing Tint 6 chicks. Therefore, establishing the nutritional requirements is fundamental and significant to achieve high performance for Jing Tint 6 chicks. We have explained this information in the Lines 51-56. Please check them, thanks.

9. Comments on the Quality of English Language: The english language must be improved and the authors must to use technical and scientific words from the field.

Response: Thank you for pointing out this. We have revised some words throughout the manuscript to make more technical and scientific. Please check them in highlighted text, thanks.

Reviewer 2 Report

Comments and Suggestions for Authors

Interesting approach to calculating nutrient requirements. A few comments and suggestions for authors in the attached pdf

Comments on the Quality of English Language

Moderate English editing might be required

Author Response

Reviewer #2: Interesting approach to calculating nutrient requirements. A few comments and suggestions for authors in the attached pdf.

1. Line 1-2: Overall the manuscript needs to be revised for English use. Maybe a native English speaker could provide some comments on the English structure and have some edits.

Response: Thank you for pointing out these. We have revised our manuscript base on your comments. We also have asked a native English speaker, who has helped us revised the English language. Please check them in highlighted text, thanks.

2. Line 46: Expert suggests a better term is “requirements” than “needs”.

Response: Thank you for pointing out this. We have changed “needs” to “requirements”.

3. Line 46: Expert suggests revising “are dynamic process” to “are a dynamic process”.

Response: Thank you for pointing out this. We have revised it as suggested.

4. Line 47: Expert suggests revising “appropriately” to “appropriate”.

Response: Thank you for pointing out this. We have changed “appropriately” to “appropriate”.

5. Line 48: Expert suggests deleting the word "production".

Response: Thank you for pointing out this. We have revised it as suggested.

6. Line 49: Expert suggests a better term is “formulation” than “preparation”.

Response: Thank you for pointing out this. We have changed “preparation” to “formulation”.

7. Line 51: Expert suggests adding “approach” after “dietary”.

Response: Thank you for pointing out this. We have revised it as suggested.

8. Line 58: Expert suggests revising “achieving” to “achieve”.

Response: Thank you for pointing out this. We have changed “achieving” to “achive”.

9. Line 58: Expert suggests deleting the word "production".

Response: Thank you for pointing out this. We have revised it as suggested.

10. Line 87: Can the authors explain this section? Do you mean the chicks were euthanized?

Response: Thank you for your question. In fact, I meant exactly what you said. We have changed “execute with ether” to “euthanized”. Please check it.

11. Line 89: Expert suggests deleting the words "of the carcasses and feathers".

Response: Thank you for pointing out this. We have revised it as suggested.

12. Line 102: Again, not sure what this is. I think you mean the chicks were euthanized. Euthanized is the correct term.

Response: Thank you for pointing out this. I apologize for my lack of precision. We have changed “executed with ether” to “euthanized”. Please check the highlighted text.

Round 2

Reviewer 1 Report

Comments and Suggestions for Authors

Thank you for the comments.